# Peptide-Based Drug Predictions for Cancer Therapy Using Deep Learning

**DOI:** 10.3390/ph15040422

**Published:** 2022-03-30

**Authors:** Yih-Yun Sun, Tzu-Tang Lin, Wen-Chih Cheng, I-Hsuan Lu, Chung-Yen Lin, Shu-Hwa Chen

**Affiliations:** 1Graduate Institute of Biomedical Electronics and Bioinformatics, College of Electrical Engineering and Computer Science, National Taiwan University, Taipei 106, Taiwan; jessie.yy.sun@gmail.com; 2Institute of Information Science, Academia Sinica, Taipei 115, Taiwan; tzutang@iis.sinica.edu.tw (T.-T.L.); wenzhi1210@gmail.com (W.-C.C.); lindalu@iis.sinica.edu.tw (I.-H.L.); 3Research Center of Cancer Translational Medicine, Taipei Medical University, Taipei 110, Taiwan

**Keywords:** anticancer peptides (ACPs), deep learning, web service, prediction

## Abstract

Anticancer peptides (ACPs) are selective and toxic to cancer cells as new anticancer drugs. Identifying new ACPs is time-consuming and expensive to evaluate all candidates’ anticancer abilities. To reduce the cost of ACP drug development, we collected the most updated ACP data to train a convolutional neural network (CNN) with a peptide sequence encoding method for initial in silico evaluation. Here we introduced PC6, a novel protein-encoding method, to convert a peptide sequence into a computational matrix, representing six physicochemical properties of each amino acid. By integrating data, encoding method, and deep learning model, we developed AI4ACP, a user-friendly web-based ACP distinguisher that can predict the anticancer property of query peptides and promote the discovery of peptides with anticancer activity. The experimental results demonstrate that AI4ACP in CNN, trained using the new ACP collection, outperforms the existing ACP predictors. The 5-fold cross-validation of AI4ACP with the new collection also showed that the model could perform at a stable level on high accuracy around 0.89 without overfitting. Using AI4ACP, users can easily accomplish an early-stage evaluation of unknown peptides and select potential candidates to test their anticancer activities quickly.

## 1. Introduction

Cell membrane properties differ between the tumor cell and healthy cells [1]. For example, the membrane fluidity of cancer cells is higher than that of healthy cells [2]. In addition, cancer cells are characterized by a negatively charged surface [3]. Anticancer peptides (ACPs), a subset of antimicrobial peptides (AMPs), are found to be toxic to cancer cells [1]. Compared with chemotherapeutic reagents used in the standard cancer treatment protocol, ACPs have higher specificity and selectivity to the neoplasm. Meanwhile, ACPs can be easily synthesized and scaled up. It can thus serve as a new option in cancer treatment modality [1].

ACPs can be divided into two types based on their putative anticancer mechanism: molecular-targeting and cancer-targeting peptides. Several state-of-the-art ACP predictors have been constructed using ACP data as a positive training set and non-ACP data as a negative training set. These predictors are helpful for scientists to evaluate peptides’ anticancer activities in anticancer agent development. However, these existing predictors were built using traditional machine learning methods. For example, a support vector machine (SVM) was applied to build AntiCP [4] and iACP [5]. Both ACPred [6] and MLACP [7] were trained using random forest (RF). Although these basic machine learning methods are popular for model construction, they have some limitations affecting model performance. Recent advances in deep learning architecture have been successfully applied in many fields (e.g., for the prediction of ACPs). For example, PTPD used Word2Vec and the deep learning network (DNN) model [8]. 

To hasten the discovery of ACPs, we built a deep learning model to detect peptides with anticancer activity. Our model was composed of a peptide sequence encoding method and a machine learning model. In this study, we used PC6 [9], a novel protein-encoding method, to convert a peptide sequence into a computational matrix, representing six physicochemical properties of each amino acid. We mainly applied the convolutional neural network to build our model. Because of an increase in the number of ACPs confirmed recently, we could identify more ACP sequences and construct a highly accurate ACP prediction model.

## 2. Results

Firstly, we recruited ACPs from publications described in the Materials and Methods. Figure 1 presents the relationship between our positive set and the dataset used in Charoenkwan et al. [10]. Data sets used in Charoenkwan et al. (to simplify the expression, we used “Charoenkwan sets”) were composed of the main data set and the alternative data set. The main data set was 861 experimentally validated ACPs as the positive set and 861 AMPs with no anticancer ability as the negative set. The alternative data set consisted of 970 ACPs as the positive set and 970 peptide sequences randomly chosen from Swiss-Prot as the negative set. Positive set in the new collection (this study, positive *n* = 2124 + negative *n* = 2124) includes Charoenkwan sets, and other 942 ACPs discovered recently. We used 80% of the data in model training and 20% for validation (Table 1) and spared one small set from the new collected ACPs (*n* = 212) as the testing set. 

We compared AI4ACP, trained using the main data set and the alternative data set in the previous study, with other ACP predictors. Most of the ACP predictors are poorly maintained, and thus they were not working or available. The results shown in Table 2 and Table 3 were obtained from the manuscripts of AntiCP2.0 [11] and ACPred [10], which were done based on Charoenkwan’s main set. As shown in Table 2, most ACP predictors trained with the main data set did not perform efficiently in low specificities or low sensitivities. 

Table 3 shows the performance of ACP predictors trained and tested using Charoenkwan’s alternative data set. AI4ACP was trained and tested in the alternative dataset to make a fair comparison. The performance of most of the ACP predictors was more favorable than those trained using the main data set. Among these predictors, AntiCP2.0 [11] exhibited the best performance. Moreover, the performance of AI4ACP trained and tested by the alternative data set is close to AntiCP2.0. 

Most ACP predictors were unable to run for the new collection due to the codes unavailable or the poor maintenance on web portals. Therefore, we picked AntiCP2.0 [11], which performed the best using the alternative dataset shown in Table 3, to compare AI4ACP through the web-based service by taking the testing set of the new collection for validation. AI4ACP was also trained using both the alternative data set and the new collection to make a fair comparison. As shown in Table 4, the performance of AI4ACP was slightly better than AntiCP2.0 when AI4ACP was trained using the alternative set. By training the model of AI4ACP in the new collection set, which is almost double the alternative set, AntiCP2.0 demonstrates the most excellent performance on the accuracy, specificity, sensitivity, and MCC in the testing set from the new collection. 

## 3. Discussions

The increasing number of publications, databases, and tools shows the importance of peptide-based therapeutics nowadays. More and more ACPs were recognized and even used as FDA-approved drugs. This result reveals the growth in demand for identifying and predicting ACPs. The identification and screening novel ACPs in a wet lab is usually time-consuming and expensive. Exploring the anticancer activity of peptides by using ACP predictors can accelerate the development of new anticancer drugs. However, the prediction of an ACP predictor is merely speculative. Laboratory experiments would still be required to confirm whether a peptide sequence possesses anticancer activity.

There are already some existing ACP predictors, such as iACP, ACPred, and AntiCP2.0. Most of the existing ACP predictors were constructed from protein-encoding methods like amino acid composition (AAC), dipeptide composition (DPC), autocovariance (AC) method, and traditional machine learning methods like SVM, random forest, or the ensemble methods. This study used a novel protein-encoding method, the PC6 protein-encoding method. The PC6 encoding method selected one property from the six subclusters of physicochemical properties of amino acids, respectively. Four of the six chosen properties were based on the seven properties from the original autocovariance (AC) methods. Moreover, two common physicochemical properties were selected from the remaining two subclusters. It was speculated that the PC6 encoding method might capture more complete features from the sequences of interest. 

The data sets used in previous studies had not been updated for quite some time. In addition to the positive data set with a few ACPs, AMPs as negative data set in previous studies might be inappropriate since ACPs are a subset of AMPs. Such poor-constructed data might reduce the accuracy of the prediction. With the up-to-date ACPs data set and an unbiased negative data set collected from UniProt and randomly generated, the predictor performed better under the same architecture.

To ensure the stability of the model and avoid overfitting during the model training, five-fold cross-validation was also applied to the model (Table 5). All the sequences used in our final model were randomly divided into five parts. In every training repetition, one of the five parts would be left out as the testing set, and the other four parts as the training set. The result showed that though the model’s performance had a slight decline as the number of sequences of the training set in five-fold cross-validation was smaller than the original training set, the average accuracy of the model is still about 89%. It showed that the model could perform at a stable level with no worries about overfitting issues.

The external testing set was used to test the model’s performance under the unknown sequences. The result showed that only 7 of 43 ACP sequences were misidentified as non-ACPs. The model’s accuracy under the external testing set is about 84%. Therefore, we could presume that if a new-designed sequence is predicted as ACP by our model, it is highly possible to be an effective anticancer peptide.

Table 2, Table 3 and Table 4 revealed that combining the PC6 encoding method and deep learning model could efficiently predict ACPs. The PC6 encoding method could exactly preserve the physicochemical properties of amino acids from original peptide sequences, and the deep learning model could learn these preserved features. In addition, with an increase in the number of peptide sequences confirmed as ACPs, we could build a predictor that exhibited more favorable performance and higher accuracy than other state-of-the-art ACP predictors. AI4ACP is a user-friendly web-based ACP predictor, and users can use this tool to detect whether the query sequence is an ACP. This tool can be beneficial for drug development for cancer treatment. AI4ACP will be continuously updated once new ACPs are discovered in the future. Besides, the deep learning model is available at https://github.com/yysun0116/AI4ACP, accessed on 6 March 2022.

## 4. Materials and Methods

### 4.1. Data Collection and Division

#### 4.1.1. Positive Data Collection

We collected ACP sequences from four ACP and AMP databases: CancerPPD [12], DBAASP [13], DRAMP [14], and YADAMP [15]. In addition, we included sequences from the positive alternative set reported by Charoenkwan et al., 2021 [10]. We downloaded all peptides with anticancer activity from the four databases and previous studies. After excluding ACPs with unusual amino acids or a nonlinear structure, namely “B”, “Z”, “U”, “X”, “J”, “O”, “i”, and “-”, and duplicates between different databases, we obtained 2839 positive ACPs. Figure 2a presents the length distribution of the 2839 ACPs; most of the sequences were shorter than 50 amino acids in length. Therefore, we excluded ACPs longer than 50 amino acids. Finally, 2815 ACP sequences were retained. Figure 2b depicts the length distribution of the 2815 ACPs. 

To ensure that the characteristics of the ACPs learned by the model were balanced, we filtered out the remaining ACPs sharing >99% sequence identity with existing ACPs by calculating the sequence identity using CD-HIT [16]. A total of 2124 ACPs were included as positive data. To evaluate the performance of our model and compare it with that of other state-of-the-art predictors, we used 10% of all the positive data as the testing set after excluding sequences from the positive set of other predictors. Figure 3 presents the detailed positive data collection and division process.

#### 4.1.2. Negative Data Collection

The negative data set consisted of 1062 non-ACP peptides from UniProt [17] and 1062 generated peptides. From UniProt, we collected peptides shorter than 50 amino acids in length and without anticancer, antiviral, antimicrobial, or antifungal activities. Random-generated peptides were derived using the same length distribution of the positive data set and randomly filled with 20 essential amino acids. Accordingly, we obtained 2124 sequences as the negative data set. We used 90% of the negative data set (1912 sequences) as the negative training set and the remaining 10% (212 sequences) as the negative testing set. Figure 4 presents the detailed negative data collection and division process.

### 4.2. Protein-Encoding Method

This study used the PC6 protein-encoding method [9] to convert a peptide sequence into a computational matrix. PC6 is a novel protein-encoding method that can encode a sequence based on both the order and physicochemical properties of the amino acids of the sequence. After benchmarking with other encoding methods, the PC6 encoding method exhibited the most satisfactory performance. Therefore, we applied PC6 in the encoding stage in our final prediction model.

### 4.3. Developing a Deep Learning Model

We implemented Keras, a high-level API from Tensorflow, to construct and train a deep learning model. We first applied the PC6 protein-encoding method [9] to all sequences. PC6 would add an extra character, “X”, which would be 0 in all six properties, at the end of the sequences for sequence padding to length 50 and convert them into 50 × 6 matrices. Figure 5 presents the process of the PC6 protein-encoding method.

Subsequently, we implemented the neural network using Keras (https://github.com/keras-team/keras, accessed on 6 March 2022) from Tensorflow2 (https://www.tensorflow.org/, accessed on 6 March 2022). The model architecture consists of three blocks composed of convolutional layers, batch normalization, max pooling, dropout layers, and two dense layers (Figure 6). The first dense layer contains 128 units with a 50% dropout rate. The last layer in the model is the output layer and is composed of a one-dimensional dense layer with the sigmoid activation function that produces a value ranging from 0 to 1; this value can indicate whether a peptide is an ACP. The convolutional layer in the three blocks in our model was built using 64, 32, and 8 one-dimensional filters of length 20 with the ReLU activation function, respectively. After the convolutional layer was built, batch normalization and max-pooling were applied with a 25% dropout rate in every block. Binary cross entropy was implemented as the loss function. With a learning rate of 0.0001, the Adam optimizer was used as our optimizer. Using the validation data set (90%), we trained the model and evaluated its performance using the validation data set (10%). Finally, all available data, namely 2124 positive and 2124 negative data, were used to train the final model.

### 4.4. Data for the Final Model

After confirming the most favorable model architecture and hyperparameters, we trained the model using all the available data (2124 positive and 2124 negative data). Eventually, we produced the final prediction model for the website. The data set used in this study can be found on our online HELP page. (https://axp.iis.sinica.edu.tw/AI4ACP/helppage.html, accessed on 6 March 2022) The positive and negative data sets will be continuously updated with the same criteria if new ACPs are discovered in the future.

### 4.5. Performance Measure

We evaluate the performance of our model using threshold-dependent parameters, which include Accuracy, Specificity, Sensitivity, and Matthews Correlation Coefficient (MCC). These parameters are calculated via the following equations:(1)$$Accuracy=\frac{TP+TN}{TP+FP+TN+FN}\times 100$$
(2)$$Specificity=\frac{TN}{TN+FP}\times 100$$
(3)$$Sensitivity=\frac{TP}{TP+FN}\times 100$$
(4)$$MCC=\frac{\left(TP\times TN\right)-\left(FP\times FN\right)}{\sqrt{\left(TP+FP\right)\left(TP+FN\right)\left(TN+FP\right)\left(TN+FN\right)}}\times 100$$
where *TP* represents the true positive predictions, *TN* represents the true negative predictions, *FP* represents the false positive predictions, and *FN* represents the false negative predictions.

### 4.6. External Testing Set

We collected ACP sequences from the updating version of DBAASP and excluded the sequences which were replicates of the 2124 ACP sequences we had collected. The sequence shorter than 10 amino acids, longer than 50 amino acids, or with amino acids out of 20 usual amino acids were also excluded. Finally, 43 ACP sequences were filtered out as an external testing set.

### 4.7. System Implementation and Workflow

For the intuitive user experience and easy understanding, we built AI4ACP composed of the LAMP system architecture (Linux Ubuntu 16.04, Apache 2.04, MySQL 5.7, and PHP 5.1) with the Bootstrap 3 CSS framework (http://getbootstrap.com/, accessed on 6 March 2022), jQuery1.11.1, and jQuery Validation version 1.17. Furthermore, the core of the analysis process was implemented in the neural network by using Keras from Tensorflow. AI4ACP runs as a virtual machine (CPU of 2.27 GHz, 20 cores, 32-GB RAM, and 500-GB storage) on the cloud infrastructure of the Institute of Information Science, Academia Sinica, Taiwan.

AI4ACP is a website service that allows users to predict whether a query peptide sequence is an ACP. The input data should be in the FASTA format, and the query peptide sequence should be composed of only 20 essential amino acids; sequences would not be recognized if they contain unusual amino acids such as B, Z, U, X, J, or O. AI4ACP would output a CSV file containing a prediction score ranging from 0 to 1 and the prediction result as YES or NO for each input peptide sequence. The prediction score represents the probability that the query peptide sequence is an ACP. The output file’s prediction results, shown as a binary column, indicate the ACP sequence(s). The prediction result is based on the prediction score with a threshold of 0.472, which is the average of thresholds calculated by training the model five times. The workflow of AI4ACP is presented in Figure 7 and explained as follows. First, the query peptide sequence is input in the FASTA format or as a FASTA file, and a valid job title is provided (Figure 7A). After the query sequence is submitted, the result appears in a three-column table composed of the input peptide’s name, prediction score, and result (Figure 7B). In addition, a pie chart presents the prediction result; this pie chart enables users to view the prediction results of the whole submission at the same time (Figure 7C).

## 5. Conclusions

ACPs are a special subset of short peptides which contain abilities to fight cancer. Modeling the ACP properties is a crucial research topic for developing ACP-based cancer therapy. This study collected up-to-date ACP data and then developed an online ACP predictor, AI4ACP. By evaluating the external testing set, our approach builds a prediction model based on the PC6 protein-encoding method, and deep learning outperforms other predictors. AI4ACP can be an ideal filter to select potential peptides in the first step of new ACP finding. Users can upload their peptide candidate sequences to our web server, get predictions in a few minutes, and pick promising ones for further costing bench experiments.

The deep learning approach in the drug discovery pipeline is beneficial for promoting and economizing the early drug development process. This study successfully transforms peptides into a machine-readable format encoded with physiochemical information. Although a large amount of data can improve model performance, it is necessary to conduct data preprocessing to prevent garbage in and out carefully. We took special care on data utilization and found that using a robust machine learning algorithm can improve model performance in learning different peptides patterns. 

## Figures and Tables

**Figure 1 pharmaceuticals-15-00422-f001:**
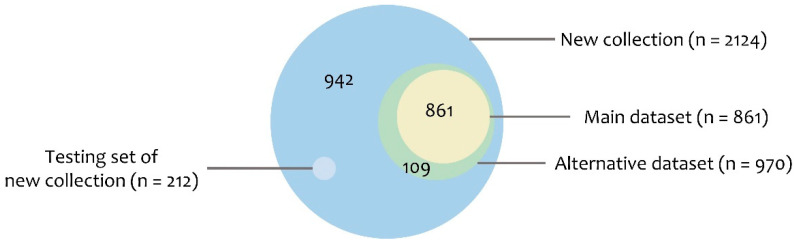
Venn diagram of the positive set of the data sets.

**Figure 2 pharmaceuticals-15-00422-f002:**
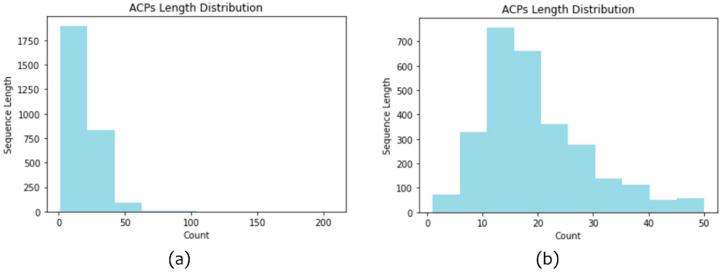
Histogram of the length distribution of collected ACPs. (**a**) The length distribution of all the 2839 ACPs; (**b**) The length distribution of ACPs, after excluding ACPs longer than 50 amino acids.

**Figure 3 pharmaceuticals-15-00422-f003:**
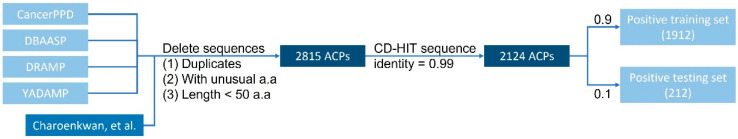
Positive data collection and division process.

**Figure 4 pharmaceuticals-15-00422-f004:**
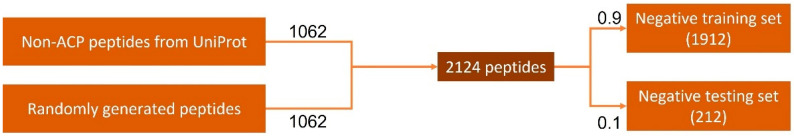
Negative data collection and division process.

**Figure 5 pharmaceuticals-15-00422-f005:**
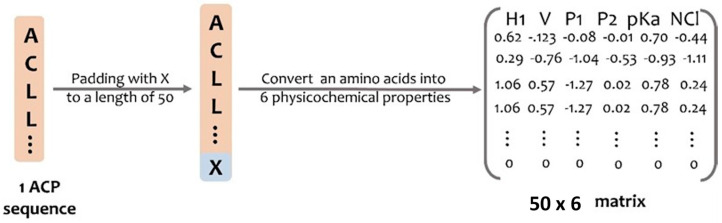
PC6 protein−encoding method. A padded ACP will be transformed into a 50 × 6 matrix.

**Figure 6 pharmaceuticals-15-00422-f006:**
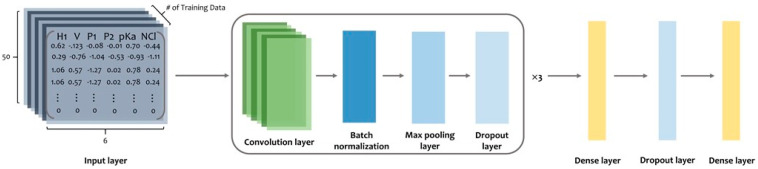
Model architecture in this study. After PC6 encoding, protein sequences will go through every layer in this model.

**Figure 7 pharmaceuticals-15-00422-f007:**
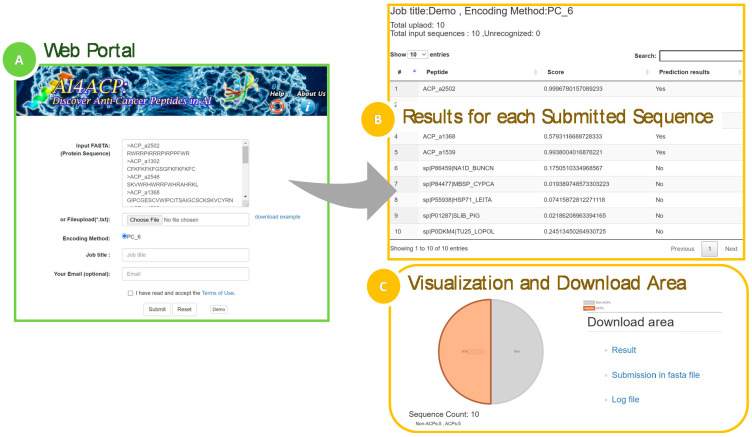
AI4ACP website. (**A**) The web portal of AI4ACP for sequence submission in FASTA. (**B**) The output of ACP activity for each submitted sequence with a prediction score. (**C**) Pie chart presents the prediction of the whole submissions and the submission with files generated during the prediction.

**Table 1 pharmaceuticals-15-00422-t001:** Comparison of the composition of three data sets.

Dataset	Dataset Usage	Positive Set	Negative Set
Main set ($P^{861}+N^{861}$)	Training set	689 ACPs	689 AMPs
	Testing set	172 ACPs	172 AMPs
Alternative set ($P^{970}+N^{970}$)	Training set	766 ACPs	776 peptides from Swiss-Prot
	Testing set	194 ACPs	194 peptides from Swiss-Prot
New collection ($P^{2124}+N^{2124}$)	Training set	1912 ACPs	956 peptides from UniProt + 956 randomly generated sequences
	Testing set	212 ACPs	106 peptides from UniProt + 106 randomly generated sequences

**Table 2 pharmaceuticals-15-00422-t002:** Comparison of ACP predictors trained and tested with the main data set. Results were obtained from AntiCP2.0 [11] and ACPred [10], except AI4ACP.

Predictors	Classifier	Accuracy	Sensitivity	Specificity	MCC *
AntiCP	SVM	0.506	**1.000** ^+^	0.012	0.070
iACP	SVM	0.551	0.779	0.322	0.110
ACPred	SVM	0.535	0.856	0.214	0.090
PEPred-Suite	ensemble approach	0.535	0.331	0.738	0.080
ACPred-FL	ensemble approach	0.448	0.671	0.225	−0.120
ACPred-Fuse	RF	0.689	0.692	0.686	0.380
AntiCP_2.0	ETree	0.754	0.775	0.734	0.510
iACP-FSCM	SVM	**0.825**	0.726	**0.903**	**0.646**
AI4ACP	CNN	0.718	0.802	0.633	0.442

*: Matthews Correlation Coefficient. ^+^: Top two ranked methods for each index are presented using text formats: first in boldface, second with underline.

**Table 3 pharmaceuticals-15-00422-t003:** Comparison of ACP predictors trained and tested using the alternative data set. Results were obtained from AntiCP2.0 [11] and ACPred [10], except AI4ACP.

Predictors	Classifier	Accuracy	Sensitivity	Specificity	MCC *
AntiCP	SVM	0.900 ^+^	0.897	0.902	0.800
iACP	SVM	0.776	0.784	0.768	0.550
ACPred	SVM	0.853	0.871	0.835	0.710
PEPred-Suite	ensemble approach	0.575	0.402	0.747	0.160
ACPred-FL	ensemble approach	0.438	0.602	0.256	−0.150
ACPred-Fuse	RF	0.789	0.644	0.933	0.600
AntiCP2.0	ETree	**0.920**	**0.923**	**0.918**	**0.840**
iACP-FSCM	SVM	0.889	0.876	0.902	0.779
AI4ACP	CNN	0.894	0.871	**0.918**	0.790

*: Matthews Correlation Coefficient. ^+^: Top two ranked methods for each index are presented using text formats: first in boldface, second with underline.

**Table 4 pharmaceuticals-15-00422-t004:** Comparison of ACP predictors tested using the testing set of the new collection.

Predictors	Classifier	Training Set	Accuracy	Specificity	Sensitivity	MCC *
AntiCP2.0	ETree	Alternative set	0.792	0.717	0.868	0.592
AI4ACP	CNN	Alternative set	0.802 ^+^	0.750	0.854	0.607
AI4ACP	CNN	New collection	**0.913**	**0.925**	**0.901**	**0.826**

*: Matthews Correlation Coefficient. ^+^: Top two ranked methods for each index are presented using text formats: first in boldface, second with underline.

**Table 5 pharmaceuticals-15-00422-t005:** Model performance of five-fold cross-validation.

Fold	Accuracy	Specificity	Sensitivity	MCC *
1	0.887	0.924	0.850	0.776
2	0.888	0.861	0.915	0.777
3	0.878	0.951	0.802	0.763
4	0.895	0.914	0.877	0.791
5	0.898	0.973	0.814	0.802
**Average**	**0.889**	**0.925**	**0.852**	**0.782**

*: Matthews Correlation Coefficient.

## Data Availability

Publicly available datasets were analyzed in this study. AI4ACP (webserver) with the dataset used is freely accessible at https://axp.iis.sinica.edu.tw/AI4ACP/, accessed on 6 March 2022. Furthermore, the model with code is also available on Github at https://github.com/yysun0116/AI4ACP, accessed on 6 March 2022.

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
