# Peer review of "Peptide-Based Drug Predictions for Cancer Therapy Using Deep Learning"

_pharmaceuticals, 2022, doi:10.3390/ph15040422_

Round 1

Reviewer 1 Report

Line 115: Please don't use ...etc expression in the sentence. Kindly rephrase it!

Line 106-157: It would be better if in the discussion section, biological implications of this benchmark effort is also devised. What would be the implication of your annotation pipeline in regard to the biomedical applications?

Line 165: Why did you exclude non-linear peptides (i.e cyclic peptides)?

Line 184: Why exclude sequences with > 50 aa?

Reviewer 2 Report

General comments:

  • Full-stop is missing in line 19.
  • Methodology should be section 2, followed by results and discussion.
  • Please give a reasoning behind technical statements. I felt, many things are stated in Abstract, Introduction, and discussion without possible background statements. This can make the manuscript very difficult to reach a wider audience, which is unfortunate as the work is very interesting.
  • The absence of conclusion and the disorder ness of the manuscript are a major red flag. I would encourage the authors to look out into this in the future.

Abstract:

  • The abstract focuses too much on the online platform and lacks technical perspective, example Deep learning is not mentioned.
  • Please specify the reason of working with the A14 ACP, and what is targets like cancer cell proliferation or inhibits migration, or suppress the formation of tumour etc.
  • Please add two or three sentences explaining what the main result reveals in direct comparison to what was thought to be the case previously, or how the main result adds to previous knowledge.
  • Please add one or two sentences to put the results into a more general context.
  • Please add two or three sentences to provide a broader perspective, readily comprehensible to a scientist in any discipline.

Introduction:

  • Line 36-40: The problem of traditional machine learning models requiring manual feature extraction may be a weak motivation to use deep learning, because there are algorithms for feature extraction like Auto encoder etc. A better way of approaching might be, while basic machine learning models do become progressively better at performing their specific functions as they take in new data, they still need some human intervention. If an AI algorithm returns an inaccurate prediction, then an engineer has to step in and make adjustments. With a deep learning model, an algorithm can determine whether or not a prediction is accurate through its own neural network—no human help is required.
  • Line 36-43: Please with the machine learning names add what kinds of data they used, what was their target and their outcomes. This should give a idea how this work differs from the other existing works.

Results:

  • This section is well explained.
  • Line 58-64: Please specify a bit more why two datasets are used.
  • Line 78-83: Suddenly here the abbreviation AMPs changes to A.M.P.s. Please keep this same throughout the manuscript.
  • Give an appropriate name for Title 2 in Table 1.
  • Please specify somewhere the actual meaning of the abbreviations Ac, Sn, Sp, MCC in Table 2, Table 3 & Table 4.
  • Can the authors clearly mention if their improved results are because of the increased and refined dataset? If so, then please provide results from SVM and Random Forest using the new dataset to clearly distinguish that Deep Learning is actually better than conventional machine learning in this scenario.

Methodology:

I am satisfied with the clarity and technical depthless of this section. Please make the other sections similar to this one.

Conclusion:

  • Please add a suitable conclusion.
  • Mdpi template provides guidelines for a good conclusion. Please follow that.

References:

  • Many references are atleast 5 years old, use recent references.

Round 2

Reviewer 2 Report

I am satisfied with the editing.